# Deep Implicit Neural Representations for End-to-End Anatomical Shape Estimation from Volumetric Images

**Max-Heinrich Laves**[1]  (iD)                                          LAVES@IMFUSION.COM
**Steffen Schuler**[1]                                                   SCHULER@IMFUSION.COM
**Ahmed Abbas**[1]                                                       ABBAS@IMFUSION.COM
**David Paik**[2]                                                        DAVID@LAZAMEDICAL.COM
**Raphael Prevost**[1]                                                   PREVOST@IMFUSION.COM
**Oliver Zettinig**[1]                                                   ZETTINIG@IMFUSION.COM
[1] *ImFusion, Munich, Germany*
[2] *Laza Medical, Campbell, USA*

**Editors:** Accepted for publication at MIDL 2025

## Abstract

We present ImplicitMeshNet[1], an end-to-end approach for anatomical shape estimation from volumetric images using deep implicit neural representations. Our neural network directly reconstructs shapes as 3D meshes and is trained on voxel-based segmentation maps by utilizing a deep signed distance field transform, eliminating the need for explicit ground truth meshes. Evaluated on cardiac CT scans from the MMWHS challenge dataset, our method achieves a Dice score of 0.92 for the extraction of the left atrium and ventricle, while maintaining anatomical fidelity. This enables more accurate cardiac modeling for visualization and downstream analysis in clinical settings.

**Keywords:** Graph convolutional network, 3D shape generation, segmentation, cardiac

## 1. Introduction

Many clinical applications require anatomical shapes represented as 3D surface meshes, including reconstructive surgery (Bauermeister et al., 2016), inter-operative visualization (Wang et al., 2021), patient-specific implant design (Mobbs et al., 2017; Chethan et al., 2019), or vascular flow simulation (Taylor et al., 2023; Saber et al., 2003). Shape estimation can be achieved by segmenting the structure of interest with deep learning techniques, followed by surface extraction using marching cubes (MC) (Lorensen and Cline, 1987). However, the resulting meshes often contain staircase artifacts, require considerable post-processing that can degrade accuracy, and are limited by the resolution of the voxel grid. This has motivated research into more direct shape estimation methods that bypass these limitations. Recent works have presented deep learning methods that directly estimate meshes from volumetric images using hybrid architectures combining a voxel encoder and a mesh decoder (Wickramasinghe et al., 2020; Kong et al., 2021). These methods rely on target meshes during training, which are typically derived from ground truth segmentation maps using MC. As MC meshes have varying topology without point correspondences to the estimated meshes, a costly and ill-posed nearest neighbor search has to be performed in every training iteration. This process is not differentiable and prevents end-to-end training

---

1. Link to public code github.com/ImFusionGmbH/ImplicitMeshNet

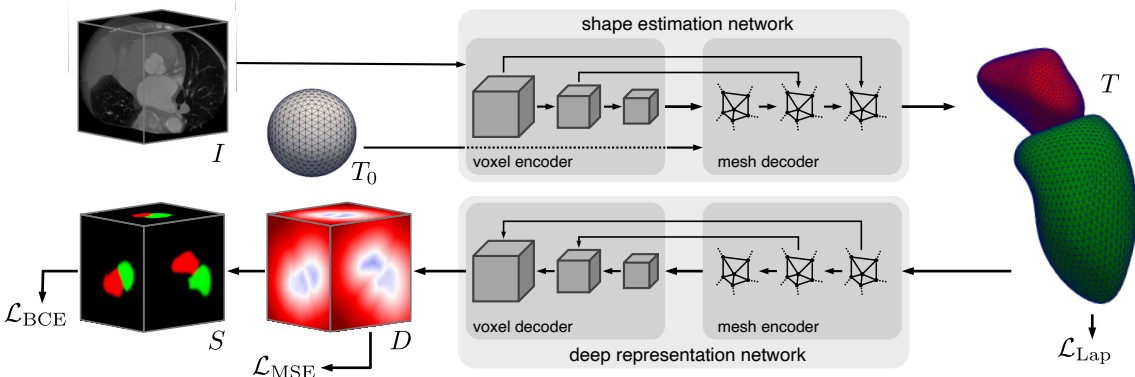

Figure 1: Overview of the proposed ImplicitMeshNet for end-to-end shape estimation.

with ground truth annotations in voxel space, for which a differentiable voxel representation of the estimated shape is needed.

To close the gap between mesh vertices and voxel images, implicit neural representations have been investigated. Methods such as Occupancy Networks (Mescheder et al., 2019) or DeepSDF (Park et al., 2019) learn to represent 3D geometry as continuous volumetric fields, such as a signed distance field (SDF), that map any point in 3D space to a scalar indicating whether this point is inside or outside of the shape. However, these methods are trained on single shapes and have to be evaluated on every coordinate in a discretized grid to derive a voxel representation, preventing their use in an end-to-end training.

## 2. Methods

We present a novel end-to-end way of training anatomical shape estimation networks on volumetric images without target meshes. Our framework consists of two parts (see Fig. 1): (1) A shape estimation network using a hybrid voxel-encoder/mesh-decoder and (2) a deep representation network mapping estimated shapes to discretized SDF on a voxel grid.

The shape estimation network $f_\theta \colon (I, T_0) \to T$ transforms the vertices $V$ of a template mesh $T_0 = (V_0, F)$ using features extracted from the input image $I$. The mesh topology defined by the set of faces $F$ is kept constant. Our network architecture is similar to Kong et al. (2021), which uses a 3D U-Net encoder and a graph convolutional decoder with Chebyshev convolutions. Vertex features are sampled from the U-Net feature maps at corresponding non-integer voxel locations using trilinear interpolation.

Our deep representation network $g_\phi \colon T \to D$ outputs a discretized signed distance field $D$ for arbitrarily transformed mesh templates. Its mesh encoder consists of repeated residual blocks using graph convolutions with ReLU activation. The decoder consists of repeated residual blocks with 3D convolutions, where each block is followed by an upsampling layer. A final residual block outputs an SDF with the same dimensions as $I$. The features from each block of the mesh encoder are projected onto the voxel grid at corresponding vertex locations using trilinear interpolation. This grid feature projection can be seen as the inverse operation of the vertex feature sampling in the shape estimation network.

The framework is trained as follows. First, we pretrain $g_\phi$ on randomly distorted template meshes $T$ using affine and elastic deformations until convergence. For each $T$, a discrete SDF $D_{\mathrm{GT}}$ is computed by a non-differentiable raytracing algorithm. The parameters $\phi$ are trained by minimizing the mean-squared error $\mathcal{L}_{\mathrm{MSE}}(D, D_{\mathrm{GT}})$ with higher weight at the zero-level set of the ground truth SDF. Next, both $f_\theta$ and $g_\phi$ are trained jointly in an end-to-end fashion using a dataset of volumetric images $I$ with corresponding binary voxel segmentations $S_{\mathrm{GT}}$. The estimated mesh $f_\phi(I, T_0) = T$ is transformed into an SDF representation $g_\phi(T) = D$, from which a label map $S$ is derived by soft binarization with a sharpened sigmoid $\sigma(-\tau^{-1}D) = S$ using a low value for temperature $\tau$. The parameters $\theta$ are optimized by minimizing the binary cross entropy $\mathcal{L}_{\mathrm{BCE}}(S, S_{\mathrm{GT}})$.

Since our approach does not employ supervision with explicit target meshes, we apply Laplacian regularization to preserve geometric consistency and ensure well-formed mesh outputs. The Laplacian loss is defined as $\mathcal{L}_{\mathrm{Lap}} = \|\mathbf{L}V\|_F^2$ where $\mathbf{L}$ is the symmetric normalized graph Laplacian matrix. This regularization promotes surface smoothness by penalizing vertices that deviate from the weighted average of their neighbors. As this is more relevant in the beginning of the training, we phase out its influence by a cosine annealed loss weight.

## 3. Results

We train our models on MMWHS (Zhuang, 2019), a public dataset containing 20 cardiac CT scans, until convergence of $\mathcal{L}_{\mathrm{BCE}}$. Two separate shape estimation networks are trained for left atrium (LA) and left ventricle (LV) segmentation, respectively. A unit sphere per structure at the center of the input image is used as template $T_0$. Affine

|  | LA | LV |
|---|---|---|
| Voxel2Mesh[2] | 0.748 | 0.669 |
| MeshDeformNet[2] | 0.926 | 0.931 |
| 3D U-Net[2] | 0.916 | 0.914 |
| ImplicitMeshNet (ours) | 0.924 | 0.921 |

Table 1: Dice scores on MMWHS test set.

and elastic deformations are used to augment the dataset. Tab. 1 shows mean Dice scores obtained from the 40 CT scans of the MMWHS test set. Qualitative results can be found in Appendix A. ImplicitMeshNet outperforms Voxel2Mesh (Wickramasinghe et al., 2020) and a 3D U-Net, while achieving comparable results to MeshDeformNet (Kong et al., 2021).

## 4. Conclusion

We presented ImplicitMeshNet, a novel end-to-end framework for anatomical shape estimation that leverages deep implicit neural representations to bridge the gap between voxel-based segmentations and surface meshes. Our approach eliminates the need for ground truth meshes during training by utilizing a deep representation network that enables direct supervision in voxel space. Initial results suggest comparable performance to state-of-the-art methods, making ImplicitMeshNet a promising alternative to existing approaches.

Future work will explore multi-organ shape estimation and extensively evaluate the method using a variety of different modalities and tasks, including more difficult shapes. The proposed framework represents a step toward more accessible and accurate high resolution vertex-based segmentation in physical space that is not limited by voxel resolution.

---

2. As reported by Kong et al. (2021)

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

## Appendix A. Additional Results

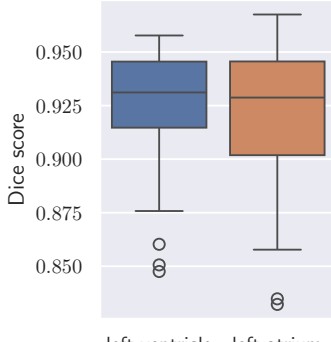

| | LA | LV |
|---|---|---|
| mean | 0.924 | 0.921 |
| median | 0.931 | 0.929 |
| std. dev. | 0.029 | 0.033 |
| min | 0.848 | 0.831 |
| max | 0.958 | 0.968 |

Figure 2: MMWHS test set: (Left) Box plots of Dice scores. (Right) Descriptive statistics.

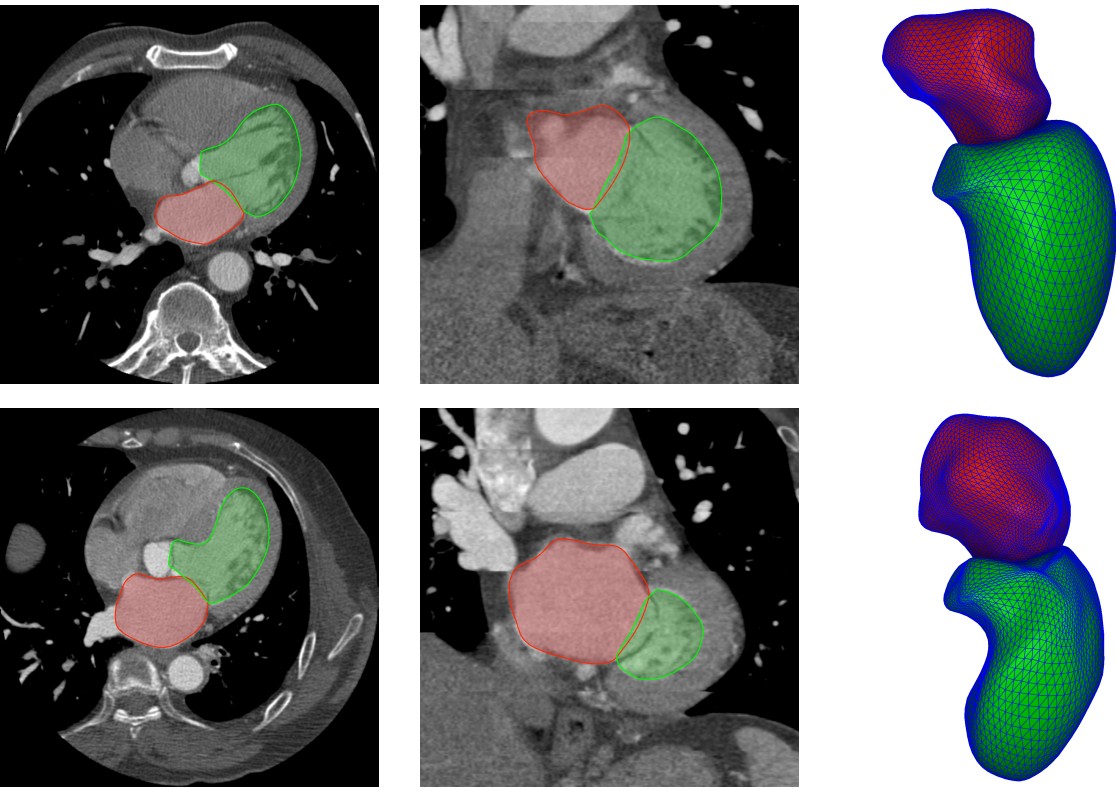

Figure 3: Qualitative results showing axial and sagittal CT views from the MMWHS test set. (Top row) Case 2001. (Bottom row) Case 2023. Slight mesh intersection can be observed as the meshes were produced by two independently trained networks.

## Appendix B. Implementation Details

PyTorch 2.6.0 (Paszke et al., 2019) and PyTorch Geometric 2.6.1 (Fey and Lenssen, 2019) were used to implement and train the networks. To ensure reproducibility, all code and hyperparameters that were used to generate the reported results are publicly available[2].

### B.1. Pretraining of DeepRepresentationNetwork

The mesh encoder of the deep representation network uses an initial graph convolution layer (Kipf and Welling, 2016) mapping the three-dimensional mesh vertices to a $C$-dimensional feature vector. This is followed by repeated residual blocks, which consists of two graph convolutions with ReLU activation layers. Dropout is used between the two convolutions to reduce overfitting. The output of each block is connected with the corresponding block of the voxel decoder using grid feature projection, where the vertex features are placed into corresponding locations on the voxel grid using trilinear interpolation (see § B.3).

The voxel decoder consists of residual blocks with 3D convolutions, group normalization, and SiLU activation functions. It progressively upsamples features using trilinear interpolation and produces multi-level predictions. During training, predictions from all levels contribute to the loss, while at inference time only the final prediction is used. For pretraining, we use synthetic data consisting of icospheres with radii ranging from 0.2 to 0.8, applying various augmentations to ensure model generalization. The augmentation pipeline includes affine transformations (rotation, scaling, translation, shearing), advanced deformations using control points with sinusoidal displacement fields, B-spline based smooth deformations, and random noise perturbations. The model is trained to predict signed distance fields from mesh vertices using MSE loss with higher weights assigned to regions near the surface. Optimization uses AdamW with weight decay and a cosine annealing learning rate schedule. This pretraining enables the deep representation network to learn a rich encoding of 3D shapes, capturing both local geometric details and global structure.

The model is trained with the following hyperparameters: learning rate of 1e-4 with cosine annealing to 1e-8, weight decay of 1e-6, and batch size of 2 over 1000 epochs. The network uses 128 hidden channels in each block with 4 stages of feature extraction and upsampling. For the training data, we generate icospheres with 4 subdivisions at various scales $(0.2-0.8)$ within $128^3$ voxel volumes.

### B.2. Training of ImplicitMeshNet

The ImplicitMeshNet architecture consists of two complementary networks: a shape deformation network $f_\theta$ that transforms a template mesh to match target anatomical structures, and the deep representation network $g_\phi$ that projects meshes back into a voxel representation, such as an SDF. Following Kong et al. (2021), we incorporate a U-Net decoder during training to provide additional segmentation supervision (Wickramasinghe et al., 2020).

The joint loss function for training the shape estimation network is:

$$\mathcal{L}_f = \alpha \mathcal{L}_{\text{BCE}} + \beta \mathcal{L}_{\text{Lap}} + \gamma \mathcal{L}_{\text{CE}} \ , \tag{1}$$

---

2. github.com/ImFusionGmbH/ImplicitMeshNet

where $\mathcal{L}_{\mathrm{BCE}}$ is the binary cross-entropy between the soft-binarized SDF and the ground truth segmentation, $\mathcal{L}_{\mathrm{Lap}}$ enforces mesh smoothness through graph Laplacian regularization, $\mathcal{L}_{\mathrm{CE}}$ is the cross-entropy loss between the U-Net decoder output and the ground truth segmentation.

We employ a cosine annealing schedule for the Laplacian regularization weight $\beta$, starting at $\beta_0 = 1000$ and decreasing to $\beta_{\mathrm{end}} = 0.01$ over training. These values were optimized via grid search to minimize training loss. This encourages initial smooth deformations while allowing more detailed surface adaptations in later epochs. The deep representation network is simultaneously trained with a weighted MSE loss as above.

For shape estimation, we use an icosphere template with radius 0.5 and 2562 vertices. Input CT scans are windowed (width=1000, level=200), normalized and resampled to $1.3\,\mathrm{mm}$ isotropic spacing with size $128^3$ using center cropping or zero padding if necessary. Training employs the AdamW optimizer with learning rates of 1e-4 for $f_\theta$ and 3e-5 for $g_\phi$, weight decay of 1e-6, batch size of 2, loss weight $\alpha = 1.0$, segmentation weight $\gamma = 0.1$, dropout probability of 0.1, and sigmoid temperature $\tau = 1\mathrm{e}\text{-}2$ for SDF binarization. The model is trained for 2000 epochs.

## B.3. Grid Feature Projection

A critical component of ImplicitMeshNet is the grid feature projection layer, which maps mesh vertices and their features back into voxel space. This bidirectional conversion between mesh and volumetric representations enables end-to-end training and consistent gradient flow through the entire network.

The grid feature projection performs trilinear interpolation of vertex features onto a regular 3D grid. For each vertex with coordinates $(x, y, z) \in [-1, 1]^3$ and associated feature vector $\mathbf{f}$, we:

1. Convert normalized vertex coordinates to voxel indices.
2. Identify the eight surrounding voxels by computing floor and ceiling indices.
3. Calculate interpolation weights based on the vertex position relative to these voxels.
4. Distribute the feature vector to each of the eight voxels, weighted by the trilinear coefficients.

For a given voxel position $\mathbf{p} = (i, j, k)$ in the grid, the feature value $\mathbf{F}(\mathbf{p})$ is computed as

$$\mathbf{F}(\mathbf{p}) = \sum_{v \in V} \mathbf{f}_v \cdot w(\mathbf{p}, \mathbf{x}_v) \ , \tag{2}$$

where $V$ is the set of all vertices, $\mathbf{f}_v$ is the feature vector of vertex $v$, $\mathbf{x}_v$ is the position of vertex $v$, and $w(\mathbf{p}, \mathbf{x}_v)$ is the trilinear interpolation weight between voxel position $\mathbf{p}$ and vertex position $\mathbf{x}_v$. The final voxel value is the accumulated sum of contributions from all vertices in the mesh. This projection mechanism is essential for enabling the neural SDF network to learn an accurate implicit representation of the deformed mesh surface.

