# OpenReview forum: "Deep Implicit Neural Representations for End-to-End Anatomical Shape Estimation from Volumetric Images"
_MIDL.io/2025/Short_Papers — MIDL 2025 - Short Papers_

### Official Review · Reviewer_ptw9 · 2025-04-27

**Rating:** 2
**Confidence:** 5

**Summary:**

This manuscript proposed an implicit network for medical shape estimation. The main idea is to build an encoder-decoder framework including CNN for volumes and GNN for meshes. Experiments on cardiac dataset demonstrated the effectiveness of the proposed method.

**Strengths:**

The writing and presentation of this manuscript is good, making it easy to read and understand. The authors have done a good job of explaining the concepts and providing examples. The encoder-decoder architecture is well-explained, and the use of this architecture for medical shape estimation also makes sense. I am also happy to see the good results on cardiac datasets based on the proposed method.

**Weaknesses:**

Although the manuscript is well-written, I have a few comments that I would like the authors to address. This method takes the volumetric data as input, and the authors should clarify how they handle the anatomies that occupies large spatial ranges. For example, the vertebrae or lung are large structures which cannot be easily handled with small patches. Without seeing the complete ROI, how can the authors ensure that the model is able to learn the shape of these structures? Besides, the experiments are only conducted on the cardiac datasets, and I would like to see more experiments on other datasets. The authors should also clarify how the proposed method can be generalized to other datasets. Finally, I would like to see a more detailed discussion of the limitations of the proposed method and how it can be improved in future work.

---

### Decision · Program_Chairs · 2025-05-01

**Decision:**

Accept

**Comment:**

The PC discussed the paper during the panel meeting and decided to accept. Given the scope of the short paper track, the PC believes this paper provides a sufficient contribution to warrant presentation at MIDL.